# DeepSleep convolutional neural network allows accurate and fast detection of sleep arousal

Hongyang Li[1] & Yuanfang Guan [1✉]

Sleep arousals are transient periods of wakefulness punctuated into sleep. Excessive sleep arousals are associated with symptoms such as sympathetic activation, non-restorative sleep, and daytime sleepiness. Currently, sleep arousals are mainly annotated by human experts through looking at 30-second epochs (recorded pages) manually, which requires considerable time and effort. Here we present a deep learning approach for automatically segmenting sleep arousal regions based on polysomnographic recordings. Leveraging a specific architecture that 'translates' input polysomnographic signals to sleep arousal labels, this algorithm ranked first in the "You Snooze, You Win" PhysioNet Challenge. We created an augmentation strategy by randomly swapping similar physiological channels, which notably improved the prediction accuracy. Our algorithm enables fast and accurate delineation of sleep arousal events at the speed of 10 seconds per sleep recording. This computational tool would greatly empower the scoring process in clinical settings and accelerate studies on the impact of arousals.

---

[1] Department of Computational Medicine and Bioinformatics, University of Michigan, 100 Washtenaw Avenue, Ann Arbor, MI 48109, USA.
✉email: gyuanfan@umich.edu

Sleep is important for our health and quality of life[1]. Inadequate sleep is often associated with negative outcomes, including obesity[2], irritability[2,3], cardiovascular dysfunction[4], hypotension[5], impaired memory[6] and depression[7]. It is estimated that around one-third of the general population in the United States are affected by insufficient sleep[8]. Spontaneous sleep arousals, defined as brief intrusions of wakefulness into sleep[9], are a common characteristic of brain activity during sleep. However, excessive arousals can lead to fragmented sleep or daytime sleepiness[2]. These arousals result from different types of potential stimuli, for example obstructive sleep apneas or hypopneas, snoring, or external noises. Currently, sleep arousals are labeled through visual inspection of polysomnographic recordings according to the American Academy of Sleep Medicine (AASM) scoring manual[10]. This is a laborious process, as the data is huge: an 8-h sleep record sampled at 200 Hz with 13 different physiological measurements contains a total of 75 million data points. It takes hours to manually score such a large-scale sleep record.

Great progress has been made in developing computational methods for automatic arousal detection based on polysomnographic recordings[11–15]. In particular, Fourier transform focusing on 30-second epochs has established one of the gold standard approaches in this field. These pioneering studies have paved the way for us to answer several important questions with the development in machine learning technologies: Which types of algorithms and data processing methods are well suited for arousal detection? How does the length of context influence the prediction outcome (i.e., input length of polysomnography record)? Which types of physiological signals should be used?

Here we investigate these questions and describe a deep learning approach, DeepSleep, for automatic detection of sleep arousals. This approach ranked first in the 2018 "You Snooze, You Win" PhysioNet/Computing in Cardiology Challenge[16], in which computational methods were systematically evaluated for predicting non-apnea sleep arousals on a large held-out test dataset[17]. The workflow of DeepSleep is schematically illustrated in Fig. 1. We built a deep convolutional neural network (CNN) to capture long-range and short-range interdependencies between time points across an entire sleep record. Information at different resolutions and scales was integrated to improve the performance. We found that similar EEG or (separately) EMG channels were interchangeable, which was used as a special augmentation in our approach. DeepSleep features fast and accurate delineation of sleep arousal events within 10 s per sleep recording. We anticipate that DeepSleep would greatly empower the scoring process in clinical settings and encourage more future studies on the impact of sleep arousals.

## Results

**Overview of the experimental design for predicting sleep arousals from polysomnogram.** In this work, we used the 994 polysomnographic records provided in the "You Snooze, You Win" PhysioNet challenge, which were collected at the Massachusetts General Hospital. In each record, 13 physiological measurements were sampled at 200 Hz (Location and Data in Fig. 1), including six electroencephalography (EEG) signals at F3-M2, F4-M1, C3-M2, C4-M1, O1-M2, and O2-M1; one electrooculography (EOG) signal at E1-M2; three electromyography (EMG) signals of chin, abdominal, and chest movements; one measurement of respiratory airflow; one measurement of oxygen saturation (SaO2); one electrocardiogram (ECG). Each time point in the polysomnographic record was labeled as "Arousal" or "Sleep" by sleep experts, excluding some non-scoring regions such as apnea or hypopnea arousals. To exploit the information of the training records, we employed a nested train–validate–test framework, in which 60% of the data was used to train the neural network, 15% of the data was used to validate for parameter selection and 25% of the data was used to evaluate the performance of the model (Cross-validation in Fig. 1). To capture the long-range and short-range information at different scales, we adapted a classic neural network (Model in Fig. 1), U-Net, which was originally designed for image segmentation[18]. Multiple data augmentation strategies, including swapping similar polysomnographic channels, were used to expand the training data space and enable the generalizability of the model. Finally, the prediction performance was evaluated by the area under receiver operating characteristic curve (AUROC) and the area under precision-recall

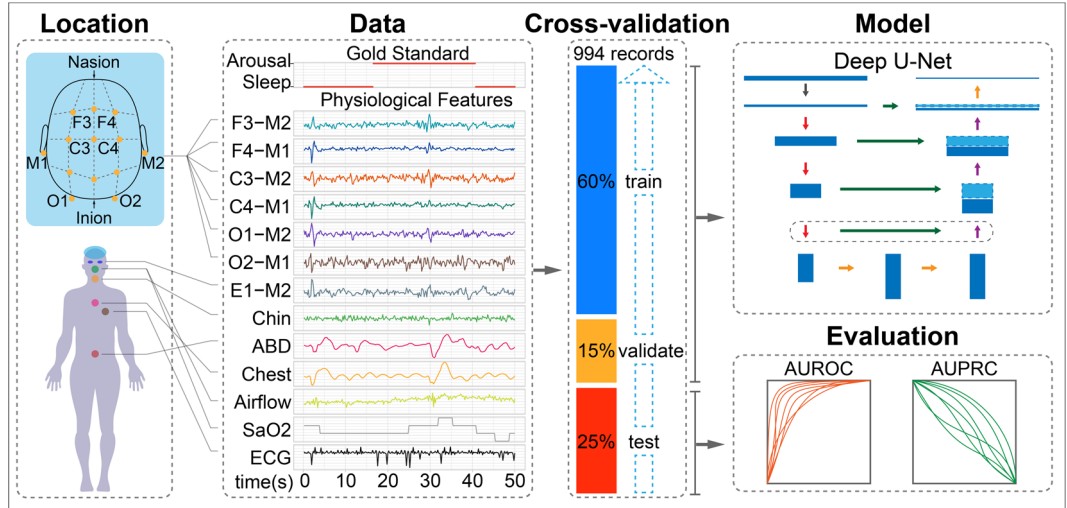

**Fig. 1 Schematic Illustration of DeepSleep workflow.** *Location*: The 13-channel polysomnogram monitored multiple body functions, including brain activity (EEG electroencephalography), eye movement (EOG, electroculography), muscle activity (EMG electromyography), and heartbeat (ECG electrocardiogram). *Data*: A 50-s sleep record example with the gold standard label of arousal/sleep and 13 physiological features. *Cross-validation*: In the nested train-validate-test framework, 60%, 15%, and 25% of the data were used to train, validate, and evaluate the model. *Model*: A 1D U-Net architecture was adapted to capture the information at different scales and allowed for detecting sleep arousals at millisecond resolution. *Evaluation*: We evaluated the predictive performance using both the area under receiver operating characteristic curve (AUROC) and area under precision-recall curve (AUPRC).

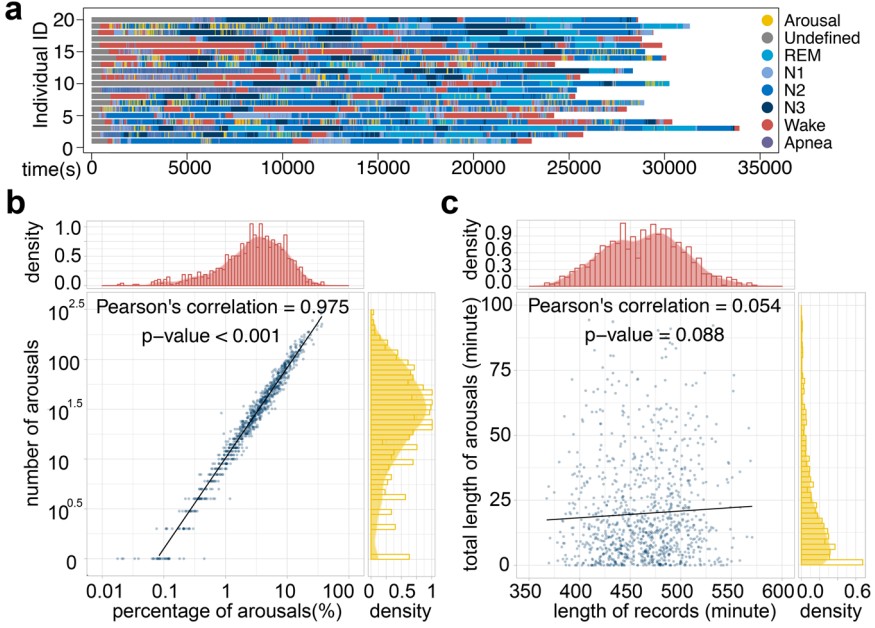

**Fig. 2 Sleep arousals sparsely and heterogeneously distributed in the sleep records. a** The eight major annotation categories are labeled in different colors for 20 randomly selected sleep records. The apneic and non-apneic arousal events happened during sleep stages (N1, N2, N3, REM). **b** The relationship is shown between the number of sleep arousals (y-axis) and the percentage of total sleep arousal time over total sleep time (x-axis) in the 994 sleep records. As expected, more arousal events are related to longer accumulated arousal time and the correlation is significantly high. **c** The length of sleep (x-axis), on the other hand, has no significant correlation with the accumulated length of sleep arousals (y-axis).

curve (AUPRC) on the held-out test dataset of 989 records (Evaluation in Fig. 1) during the challenge.

**Heterogeneous sleep arousal events among individuals challenge the development of automatic scoring method.** Compared with automatic sleep stage detection, scoring sleep arousals are more challenging owing to the fact that arousal events are commonly heterogeneously and sparsely distributed during sleep among different individuals. We investigated the annotations of these sleep records and found high levels of heterogeneity among individuals. In Fig. 2a, we randomly selected sleep records of 20 individuals and presented the annotations in different colors. There are eight major annotation categories: "Arousal", "Undefined", "REM" (Rapid Eye Movement), "N1" (Non-REM stage 1), "N2" (Non-REM stage 2), "N3" (Non-REM stage 3), "Wake" and "Apnea". The distribution of these categories differs dramatically among individuals (different colors in Fig. 2a). Clearly, different individuals display distinct patterns of sleep, including the length of total sleep time and multiple sleep stages. Notably, the sleep arousal regions are relatively short and sparsely distributed along the entire record for most individuals (yellow regions in Fig. 2a).

We further investigated the occurrence of arousals and found that the median number of non-apneic arousals was 29 during the entire night of recording. A total of 82 individuals (8.25%) had more than 100 non-apneic arousals during their sleep (y-axis in Fig. 2b), lasting around 10% of the total sleep duration (x-axis in Fig. 2b). In addition, there was no significant correlation between the total sleep time and the total length of sleep arousals (Fig. 2c), which was expected since the quality of sleep is not determined by sleep length. In summary, the intrinsically high heterogeneity of sleep records across individuals rendered the segmentation of sleep arousals a very difficult problem.

**Deep U-Net captures the long-range and short-range information at different scales and resolutions.** Current manual annotation of sleep arousals is defined by the AASM scoring

manual[10], in which sleep experts focus on a short period (less than a minute) and make decisions about sleep arousal events. However, it remains unclear whether the determinants of sleep arousals reside only within a short range, or long-range information across minutes and even hours plays an indispensable role in detecting sleep arousals. Although sleep arousal is in nature a transient event, it may be associated with the overall sleep pattern through the night. We tested input lengths ranging from 4096 time points (~20 s) to 131,072 time points (~10 min), with single-channel input (Supplementary Fig. 1a, b) or all-channel input (Supplementary Fig. 1c, d). Intriguingly, when we trained the convolutional neural networks on longer sleep records, we consistently achieved better performances. Therefore, we used the entire sleep record as input to make predictions, instead of small segments of a sleep record.

To learn the long-range association between data points across different time scales (second, minute, and hours), we develop an extremely deep convolutional neural network, which contains a total of 35 convolutional layers (Fig. 3a). This network architecture has two major components, the encoder and the decoder. The encoder takes a full-length sleep record of $2^{23} = 8,388,608$ time points and gradually encrypts the information into a latent space (the red trapezoid in Fig. 3a). Sleep recordings were centered, regardless of their original lengths, within the 8-million input space by filling in with zeros on their extremes. To be specific, the convolution–convolution-pooling (hereafter referred to as "ccp") block is used to gradually reduce the size from $2^{23} = 8,388,608$ to $2^8 = 256$ (Fig. 3b top). Meanwhile, the number of channels gradually increases from 13 to 480 to encode more information, compensating the loss of resolution in the time domain. In each convolutional layer, the convolution operation is applied on the data along the time axis to aggregate the neighborhood information. Since the sizes of data in these convolutional layers are different, the encoded information is unique within each layer. For example, in the input layer, 10 successive time points sampled at 200 Hz correspond to a short time interval of $10/200 = 0.05$ s, whereas in the center layer

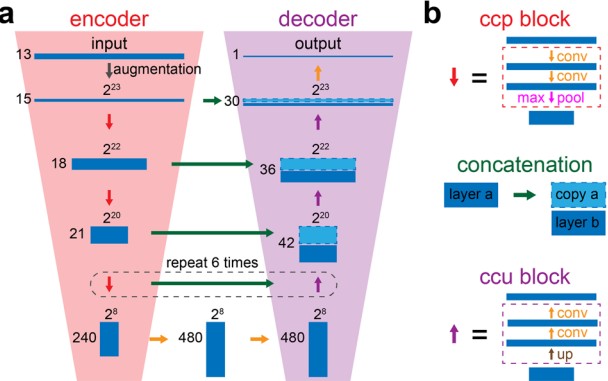

**Fig. 3 The deep convolutional neural network architecture in DeepSleep.**
**a** The classic U-Net structure was adapted in DeepSleep, which has two major components of the encoder (the red trapezoid on the left) and the decoder (the purple trapezoid on the right). **b** The building blocks of DeepSleep are the convolution–convolution-pooling block (red), the concatenation (green), and the convolution–convolution-upscaling block (purple). The orange arrow represents the convolution operation.

(size $= 2^8$), 10 time points correspond to a much longer time interval of $0.05*2^{23-8} = 1638$ s, nearly 30 min. Therefore, this deep encoder architecture allows us to capture and learn about the interactions across data points at multiple time scales. The relationship between length of segments and the corresponding time can be found in Supplementary Table 1.

Similar to the encoder, the second component of our network architecture is a decoder to decrypt the compressed information from the center latent space. In contrast to the "ccp" block, the convolution–convolution-upscaling (hereafter referred to as "ccu") block is used (Fig. 3b bottom), which gradually increases the size and decreases the number of channels of the data (the purple trapezoid in Fig. 3a). In addition, concatenation is used to integrate the information from both the encoder and the decoder at each time scale (green horizontal arrows in Fig. 3). Finally, the output is the segmentation of the entire sleep record, where high prediction values indicate sleep arousal events and low values indicate sleep.

**Deep learning enables accurate predictions of sleep arousals**. By capturing the information at multiple resolutions, DeepSleep achieves high performance in automatic segmentation of sleep arousals. Since deep neural networks are iteration-based machine learning approaches, a validation subset is used for monitoring the underfitting or overfitting status of a model and approximating the generalization ability on unseen datasets. A subset of 15% randomly selected records was used as the validation set during the training process (Cross-validation in Fig. 1) and the cross entropy was used to measure the training and validation losses (see details in "Methods" section). We developed three basic models called "1/8", "1/2", and "full", according to the resolution of the neural network input. The "full" resolution means that the original 8-million ($2^{23} = 8,388,608$) length data were used as input. The "1/2" or "1/8" resolution means that the original input data were first shrunk to the length of 4-million ($2^{22}$) or 1-million ($2^{20}$) by averaging every 2 or 8 successive time points, respectively. We observed similar validation losses of the "full", "1/2", and "1/8" models (solid lines in Fig. 4a). The final evaluation was based on the AUROC and AUPRC scores of predicting 25% of the data. In Fig. 4b, each blue dot represented one sleep record and we observed a significant yet weak correlation = 0.308 between the AUROCs and AUPRCs. The baselines of random predictions were shown as red dashed lines. Notably,

the AUPRC baseline of 0.072 corresponded to the ratio of the average total sleep arousal length over the total sleep time, which was considerably low and made it a hard task due to the intrinsic sparsity of sleep arousal events.

To build a robust and generalizable model, multiple data augmentation strategies were used in DeepSleep. After carefully examining the data, we found that signals belonging to the same physiological categories were very similar and synchronized, including two EMG channels and six EEG channels (see Data in Fig. 1). We applied an augmentation strategy by randomly swapping these similar channels during the model training process, assuming that these signals were interchangeable in determining sleeping arousals. There are three EMG channels but EMG-chin were not considered in this swapping strategy due to its differences from the other two EMG (ABD and chest) channels (see Data in Fig. 1). This channel swapping strategy was bold but effective, adapting which largely improved the prediction performance ("1/8_no_swap" versus "1/8" evaluated by the AUPRC, AUROC, Sorensen dice coefficient, and Jaccard index in Fig. 4c–f). Finally, we assembled the predictions from the "1/8", "1/2", and "full" resolution models as the final prediction in DeepSleep ("1/8 + 1/2 + full" in Fig. 4c–f). We further extended our algorithm and trained a multi-task model using the sleep stage scores as the ground truth labels, including REM, N1, N2, N3, Wake. Similar to sleep arousal detection, we evaluated the predictive performance of our model using cross validations (Fig. 4g, h). In general, this multi-task model achieved high AUROCs and AUPRCs, demonstrating the robustness and generalizability of our deep learning pipeline in multiple sleep staging.

We further investigated which types of physiological signals are necessary for sleep arousal detections and benchmarked the performance of (1) models using all channels and (2) models using one type of signals (EEG, EOG, EMG, Airflow, Saturation of Oxygen, or ECG). The results are shown in Fig. 5a, b. We found that models with only EMG achieved relatively high performance (AUPRC = 0.476, AUROC = 0.902), close to the model with all channels (AUPRC = 0.520, AUROC = 0.919). For models with other types of channels, the AUPRCs and AUROCs are around 0.3 and 0.8, respectively. The 13 polysomnographic channels complemented each other and using all of them instead of one type of signals allowed the neural network to capture interactions between channels and achieved the highest performance. In addition, we multiplied the polysomnographic signals by a random number between 0.90 and 1.15 to simulate the inherent fluctuation and noise of the data. Other augmentations on the magnitude and time scale were also explored (Fig. 5c, d). Furthermore, to address the heterogeneity and batch effects among individuals, we quantile normalized each sleep record to a reference, which was generated by averaging all the records. This step effectively removed the biases introduced by the differences of individuals and instruments, and Gaussian normalization was also tested and had slightly lower performance (Fig. 5e, f).

We further compared different machine learning models and strategies in segmenting sleep arousals. We first tested a classical model, logistic regression, and found that our deep learning approach had a much higher performance (Fig. 5g, h). It has also been reported that neural network approaches outperformed classical machine learning methods, including random forest, logistic regression[19], support vector machine, and linear models[20]. In fact, 8 out of the top 10 teams used neural network models in the PhysioNet Challenge (red blocks in Supplementary Fig. 2a)[16]. Different input lengths and data preprocessing strategies were used, including raw polysomnogram data, features extracted by statistical analysis, short-time Fourier transform, or wavelet transform. Two types of network structures (convolutional and recurrent) were mainly used, and integrating Long

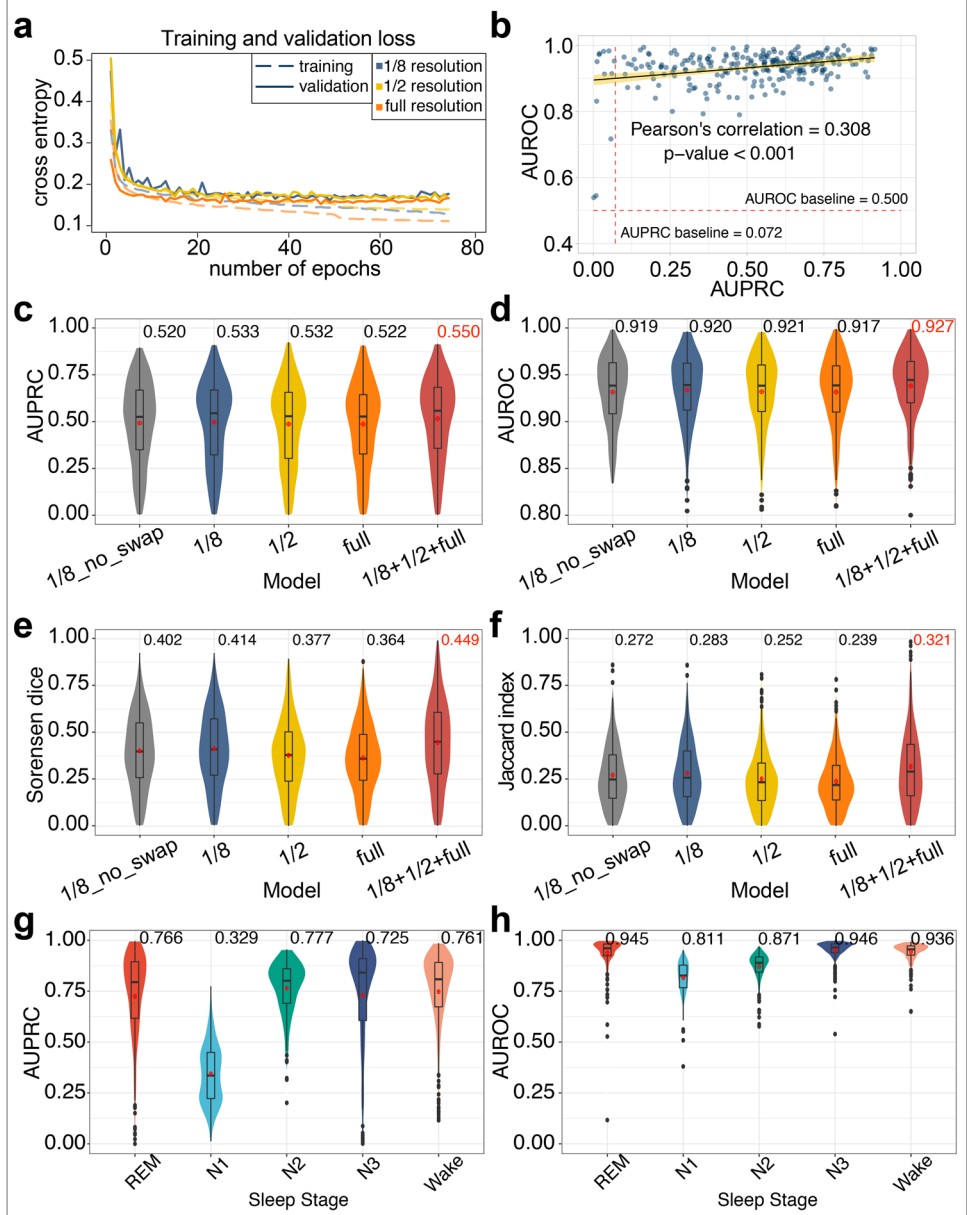

**Fig. 4 The performance comparison of DeepSleep using different model training strategies. a** The training and validation cross entropy losses are shown in the dashed and solid lines, respectively. The models using sleep records at different resolutions are shown in different colors. **b** The prediction of each sleep record in the test set is shown as a blue dot in the area under receiver operating characteristic curve (AUROC)–area under precision-recall curve (AUPRC) space. A weak correlation is observed between AUROCs and AUPRCs with a significant $p$-value < 0.001 on the $n = 261$ test records. The 95% percent confidence interval is shown as the yellow bend. The baselines of random predictions are shown as red dashed lines. The prediction **c** AUPRCs, **d** AUROCs, **e** Sorensen dice coefficient, and **f** Jaccard index of models using different resolutions or strategies were calculated. The "1/8_no_swap" model corresponds to the model using the "1/8" resolution records as input without any channel swapping, whereas the "1/8", "1/2", and "full" models use the strategy of swapping similar polysomnographic channels. The final "1/8 + 1/2 + full" model of DeepSleep is the ensemble of models at three different resolutions, achieving the highest AUPRC of 0.550 and AUROC of 0.927. In addition to predicting sleep arousal, we also extended our method for sleep staging (REM, N1, N2, N3, and Wake) and evaluated performance using (**g**) AUPRCs and (**h**) AUROCs.

Short-Term Memory (LSTM) or Gated Recurrent Unit (GRU) into DeepSleep did not improve the performance (Supplementary Fig. 2b–d). In terms of input length, increasing input length considerably improved the performance, and full-length records were used by three teams (blue blocks in Supplementary Fig. 2a). We also compared DeepSleep with recent state-of-the-art methods in sleep stage scoring. These methods extracted features from 30-s epochs through short-time Fourier transform (STFT)[21,22] or Thomson's multitaper[19,23]. They were originally designed for automatic sleep staging and we applied them to the task of detecting sleep arousals on the same 2018 PhysioNet data. Although these methods performed well in sleep stage scoring, they were not well suited for arousal detection (Supplementary Figure 2e, f). Deep learning approaches can model informative features in an implicit way without tedious feature crafting[24], and neural networks using raw data as input were frequently used by half of the top 10 teams (orange blocks in Supplementary Fig. 2a).

To comprehensively investigate the effects of various network structures and parameters on predictions, we further performed experiments with different modifications, including (1) the

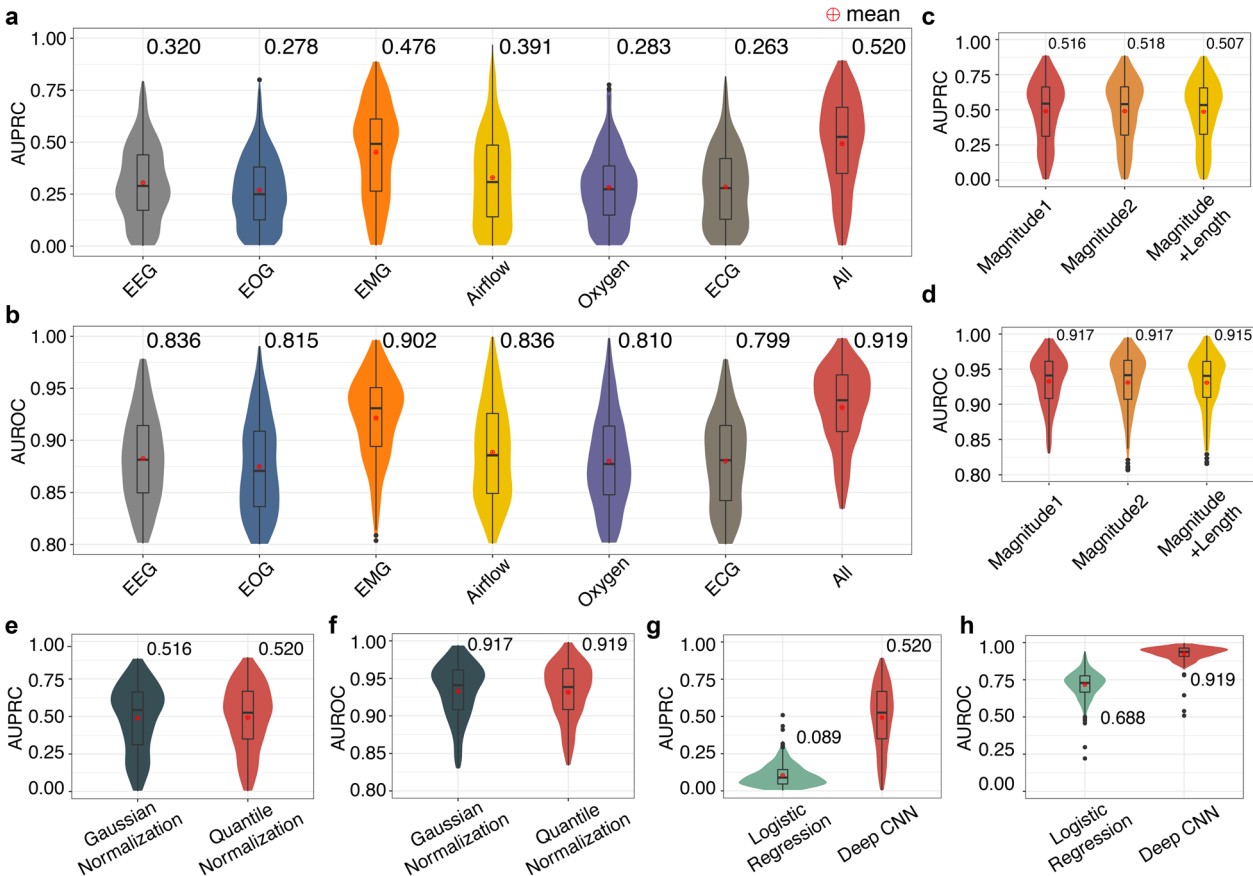

**Fig. 5 The performance comparison of models using different types of polysomnographic signals, augmentation strategies, normalization methods.**
From left to right, the first six categories are EEG (channels 1–6), EOG (channel 7), EMG (channels 8–10), Airflow (channel 11), saturation of oxygen (channel 12), and ECG (channel 13). The last one, "All", represents the model using all these 13 channels as input. The prediction **a** AUPRCs (area under precision-recall curve) and **b** area under receiver operating characteristic curve (AUROCs) of models using different types of signals are shown in different colors. The model "All" using all 13 polysomnographic signals achieved the best performance. We further compared the prediction **c** AUPRCs and **d** AUROCs of different data augmentation strategies. The "Magnitude 1" strategy means that each training record was multiplied by a random number between 0.90 and 1.15, to simulate the fluctuation of the measurement in real life. The "Magnitude 2" strategy was the same as "Magnitude 1", except for the range of the random number becoming wider, between 0.80 and 1.25. These two strategies had almost the same performance. The last "Magnitude + Length" strategy was built on top of "Magnitude 1", in which we further extended or shrunk the record along the time axis by a random number between 0.90 and 1.15. This strategy decreased the performance and was not used in the final model training. In addition, the prediction **e** AUPRCs and **f** AUROCs of the Gaussian normalization and the quantile normalization were compared. In the Gaussian normalization, we first subtracted the average value of a signal then divided the signal by the standard deviation for each sleep record. In the quantile normalization, we first calculated the average of all training records as the reference record. Then for each record, we quantile normalized it to the reference record. The quantile normalization had better performance. We also compared the prediction **g** AUPRCs and **h** AUROCs of the deep convolutional neural network (CNN) and logistic regression. The value above each violin is the overall AUPRC/AUROC, which is different from the simple mean or median value. The overall AUPRC/AUROC considers the length of each record and longer records contribute more to the overall AUPRC/AUROC (see details in "Methods" subsection "Overall AUPRC and AUROC").

"shallow" neural network with less convolutional layers (Supplementary Fig. 3a, b), (2) using average pooling instead of max pooling (Supplementary Fig. 3c, d), (3) larger convolution kernel size from 7 to 11 (Supplementary Fig. 3e, f), and (4) using the Sorensen dice loss function instead of cross-entropy loss (Supplementary Fig. 3g, h). These modifications had either similar or lower prediction performances. We concluded that the neural network architecture and augmentation strategies in DeepSleep were optimized for the current task of segmenting sleep arousals. Subsequent analysis of the relationships between the predictive performance and the number of arousals were performed (Supplementary Fig. 4a, b). As we expected, the prediction AUPRC was positively correlated with the number of arousals in a sleep record. The individuals who had more sleep arousals during sleep were relatively easier to predict. As a control, we also calculated Pearson's correlations between AUPRC/AUROC and the total length of sleep record

(Supplementary Fig. 4c, d). These correlations are close to zeros and not statistically significant, with the $p$-values of 0.176 and 0.316, respectively ($n = 261$ test records). Moreover, we tested the runtime of DeepSleep with Graphics Processing Unit (GPU) acceleration and segmenting sleep arousals of a full sleep record can be finished within 10 s on average (Supplementary Fig. 4e, f). The time cost of DeepSleep is much lower than that of manual annotations, which requires hours for one sleep record.

In addition to the 2018 PhysioNet dataset, we further validated our method on the large publicly available Sleep Heart Health Study (SHHS) dataset, which contains 6441 individuals in SHHS visit 1 (SHHS1) and 3295 individuals in SHHS visit 2 (SHHS2)[25]. The SHHS is a multi-center cohort study, including participants from multiple different cohorts and the polysomnograms were annotated by sleep experts from different labs (https://sleepdata. org/datasets/shhs). The recording montages and signal sampling rates of SHHS1 and SHHS2 were quite different. For both SHHS1

and SHHS2, we randomly selected 1000 recordings, which was comparable to the number of recordings ($n = 994$) in the

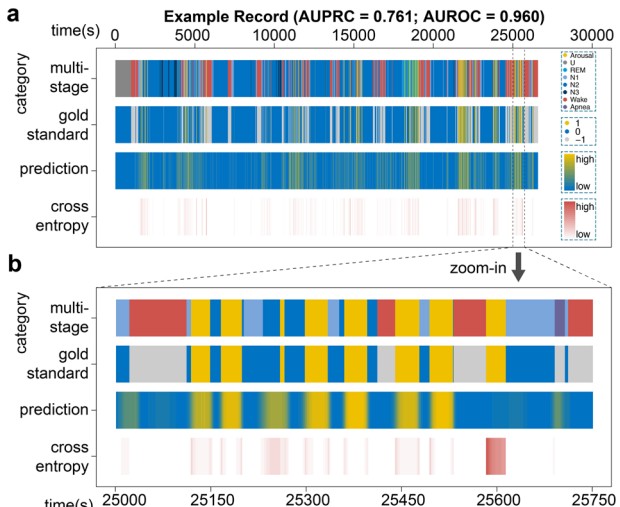

**Fig. 6 Visualization of DeepSleep predictions and the gold standard annotations. a** A 7.5-h sleep record (id = tr05−1034) with the prediction area under receiver operating characteristic curve (AUROC) of 0.960 and area under precision-recall curve (AUPRC) of 0.761 is used as an example. From top to bottom along the y-axis, the four rows correspond to the eight annotation categories, the binary label of arousal (yellow), sleep (blue), and the non-scoring regions (gray), the continuous prediction, and the cross-entropy loss at each time point along the x-axis. The wrongly predicted regions lead to high cross-entropy losses, which are shown in dark red at the bottom row. **b** The zoomed in comparison of a 12.5-min period of this sleep record.

PhysioNet training dataset. Then we applied DeepSleep pipeline to train, validate, and test models on SHHS1 and SHHS2 datasets individually. We observed similar performances of detecting sleep arousals on the PhysioNet, SHHS1, and SHHS2 datasets in Supplementary Fig. 5a, b, demonstrating the robustness of our method. We also summarized the available clinical characteristics of the PhysioNet, SHHS1, and SHHS2 datasets in Supplementary Tables 2–4, including gender, age, race, and disease. The different groups were balanced for the training and testing subsets. We further evaluated the predictive performance for these groups (Supplementary Figs. 5c–h and 6). In general, the performances were very similar among different groups with the AUPRCs around 0.6. We found slightly higher performance for males than females in all three datasets (Supplementary Fig. 5c–e). There are some differences for different age groups, but no clear trends were observed (Supplementary Fig. 5f–h). In terms of race, we found slightly higher predictive performance for the others and white groups than the black group (Supplementary Fig. 6a, b). The prediction AUPRC is highly associated with the AUPRC baseline, which is the ratio of sleep arousal time over the total sleep time. For different gender and race groups, the main reason for a higher AUPRC was that the corresponding baseline was higher (Supplementary Table 5). For example, on average males commonly had more sleep arousals (higher AUPRC baselines) than females in three datasets. As we expected, our model achieved higher AUPRCs for males. For the SHHS1 and SHHS2 datasets, we also considered patients with different cardiovascular diseases/events, including myocardial infarction (MI), stroke, angina, and congestive heart failure (CHF). The predictive performances of different groups were similar, with the AUPRCs ranging from 0.56 to 0.61 (Supplementary Fig. 6c, d).

In the clinical setting, both apneic and non-apneic arousals are important. We have therefore built neural network models for detecting apnea, in addition to the model for detecting

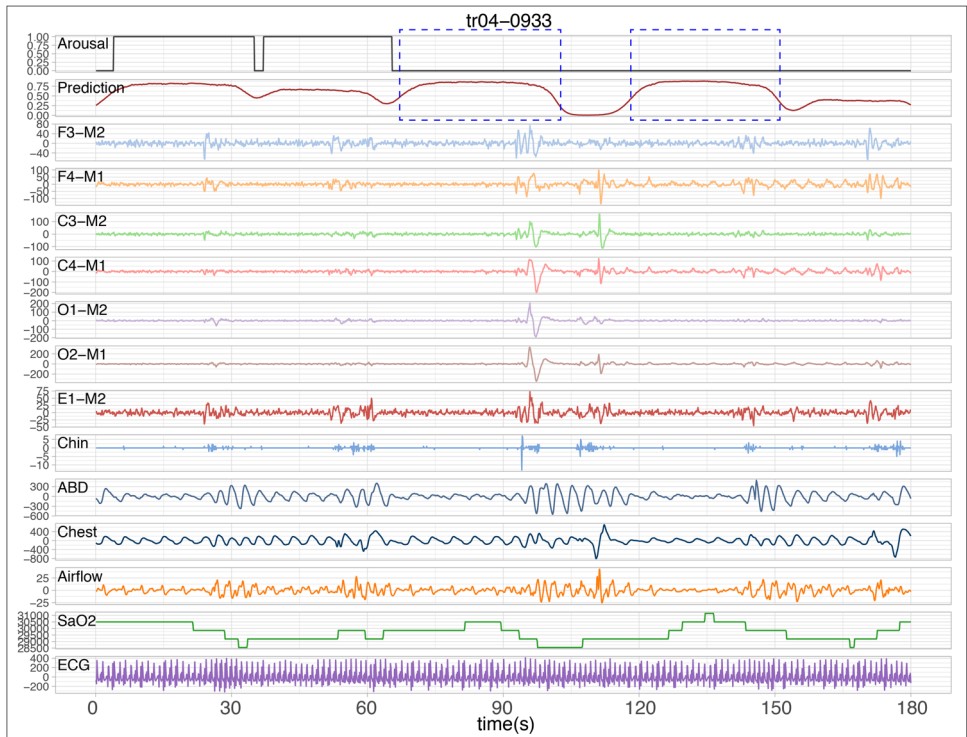

**Fig. 7 A 180-s polysomnogram example with manual labels and predictions of sleep arousals.** From top to bottom, the sleep arousal labels (arousal = 1 and non-arousal = 0), predictions by our algorithm, and 13-channel polysomnograms are shown for a total of 180 s (six 30-s epochs). In addition to the two arousal events at the very beginning, our algorithm also detected two suspected sleep arousal events shown in dashed blue rectangles.

non-apneic arousals, which was originally designed during the 2018 PhysioNet challenge. Specifically, we applied DeepSleep pipeline to the PhysioNet data and built three types of models for detecting (1) apneic arousals; (2) non-apneic arousals; and (3) all arousals (apneic and non-apneic arousals). DeepSleep is able to detect both apneic and non-apneic arousals (Supplementary Fig. 7a, b). We further explored the potential usage of feature maps after the encoder blocks in our neural network. We trained a lightGBM model to predict other sleep characteristics, including apnea–hypopnea index (AHI) and percentages of N1/N2/N3/REM. We observed medium Pearson's correlations between predictions and observations (Supplementary Fig. 7c), indicating that the latent feature maps can be also used in sleep related studies.

**Visualization of DeepSleep predictions**. In addition to the abstract AUROC and AUPRC scores, we directly visualized the prediction performance of DeepSeep at 5-ms resolution (corresponding to the 200 Hz sample rate). An example 7.5-h sleep record with the prediction AUROC of 0.960 and AUPRC of 0.761 is shown in Fig. 6. More examples at 3 rank percentiles (25%, 50%, and 75%) based on the AUPRC values can be found in Supplementary Fig. 8. From top to bottom, we plotted the multi-stage annotations, sleep arousal labels, predictions and cross-entropy losses along the time x-axis. By comparing the prediction and gold standard, we can see the general prediction pattern of DeepSleep correlates well with the gold standard across the entire record (the second and third rows in Fig. 6a). We further zoom into a short interval of 12.5 min and DeepSleep successfully identifies and segments seven sleep arousal events out of eight (yellow in Fig. 6b), although one arousal around 25,600 is missed. Intriguingly, DeepSleep predictions display a different pattern from the gold standard annotated by sleep experts: DeepSleep assigns continuous prediction values with lower probabilities near the arousal–sleep boundaries, whereas the gold standard is strictly binary either arousal = 1 or sleep = 0 based on the AASM scoring manual[10]. This becomes clearer when examining the cross entropy loss at each time point and the boundary region has higher losses shown in red (the bottom row in Fig. 6b). This is expected because in general we will have a higher confidence of annotation in the central region of sleep arousal or other sleep events. Yet due to the limit of time and effort, it is practically infeasible to introduce rules for manually annotating each time point via a probability scenario. Additionally, binary annotation of sleep records containing millions of data points has already required considerable effort. DeepSleep opens a avenue to reconsider the way of defining sleep arousals or other sleep stage annotations by introducing the probability system.

It is critical to compare predictions of our algorithm and the ground truth labels created by sleep scorers, since the manual scores may not be perfect. Specifically, we focused on the false positives detected by our method and added five examples from five different individuals in Fig. 7 and Supplementary Figs. 9–12. In each figure, the ground truth arousal labels created by human scorers (arousal = 1 and non-arousal = 0) and our predictions are shown in the top two rows. The 13 physiological channels from the polysomnogram are also shown below. A total of six 30-s epochs are shown in each figure, containing multiple true positives and false positives (dashed blue rectangles). For example, in Fig. 7 (tr04-0933), there are two sleep arousal events detected by both manual scorers and our algorithm. Our algorithm also found two other suspected sleep arousal events shown in dashed blue rectangles. By examining the corresponding polysomnographic signals, we found that these two false positives were likely to be sleep arousals missed by the scorers—there are

sudden shifts in the EEG and EOG channels, and suspected respiratory effort related changes in the EMG channels (ABD and Chest) and Airflow. Similarly, we identified multiple false positives for other individuals in Supplementary Figs. 9–12. These results indicate that our computational method can potentially detect suspected arousals, complementing the scoring by human experts, and highlighting the regions of interest to assist sleep scoring.

## Discussion

A major challenge in research on sleep arousals is that the process of arousal detection is tedious and sometimes unreliable. A fast and accurate computational tool would greatly empower the scoring process and accelerate studies on the impact of arousals. In this work, we created a deep learning approach, DeepSleep, to automatically segment sleep arousal regions in a sleep record based on the corresponding polysomnographic signals. A deep convolutional neural network architecture was designed to capture the long-range and short-range interactions between data points at different time scales and resolutions. Unlike classical machine learning models[26], deep learning approaches do not depend on manually crafted features and can automatically extract information from large datasets in an implicit way[27]. Using classical approaches to define rules and craft features for modeling sleep problems in real life would become much too tedious. In contrast, without assumptions and restrictions, deep neural networks can approximate complex mathematical functions and models to address those problems. Currently, these powerful tools have also been successfully applied to biomedical image analysis and signal processing[28,29]. Compared with classical machine learning models, deep learning is a "black box" method which is relatively hard to interpret and understand. Meanwhile, deep learning approaches usually require more computational resources such as GPUs, whereas most classical machine learning models can run on common CPUs.

Overfitting is a common issue in deep learning models, especially when the training dataset is small and the model is complex. Even if we use a large dataset and perform cross-validation, we will gradually and eventually overfit to the data. This is because each time we evaluate a model using the internal test set, we probe the dataset and fit our model to it. In contrast to previous studies, the 2018 PhysioNet Challenge offered us a unique opportunity to truly evaluate the performances and compare cutting-edge methods on a large external hidden test set of 989 samples[17]. In addition, we demonstrate that deep convolutional neural networks trained on full-length records and multiple physiological channels have the best performance in detecting sleep arousals, which are quite different from pioneering approaches extracting features from short 30-s epochs[19,21,24]. Beyond sleep arousals, we propose that the U-Net architecture used in DeepSleep can be adapted to other segmentation tasks such as sleep staging. A multi-tasking learning approach can be further implemented as the outputs of U-Net to directly segment multiple sleep stages simultaneously based on polysomnograms.

An interesting observation is that when we used records of different lengths as input to train deep learning models, the model using full-length records largely outperformed models using short periods of records. This observation brings about the question of how to accurately detect sleep arousals based on polysomnography. Current standards mainly focus on short time intervals of less than one minute[10], yet the segmentations among different sleep experts are not very consistent in determining sleep arousals. One reason is that it is hard for humans to directly read and process millions of data points at once. In contrast, computers are good at processing large-scale data and discovering

the intricate interactions and structures between data points across seconds, minutes and even hours. Our results indicate that sleep arousal events are not solely determined by the local physiological signals but associated with much longer time intervals even spanning hours. It would be interesting to foresee the integration of computer-assisted annotations to improve definitions of sleep arousals or other sleep stages.

In addition to the unique long-range information captured by DeepSleep, a clear advantage of computational approaches lies in the annotations for the boundary regions between arousal and sleep. Since current sleep annotations are binary only, it would be a more accurate and appropriate approach to introduce the probability of the annotation confidence, especially at the boundary regions. Machine learning approaches such as DeepSleep naturally provide continuous predictions for each time point. It would be interesting to see improved annotation systems using continuous values instead of binary labels. A simple approach could be directly integrating the computer predictions with annotations by human sleep experts. The proposed annotation systems would provide more accurate information for the assessment of patients for sleep disorders and the evaluation of sleep quality in the future.

## Methods

**Polysomnographic recordings**. The dataset used in this study contains a total of 994 polysomnographic sleep records from different individuals and their corresponding labels at each time point. Specifically, the arousal region is labeled by "1" and other sleep regions are labeled by "0", except for the wakefulness regions, apnea arousal regions, and hypopnea arousal regions labeled by "−1". These "−1" regions will not be scored in the challenge, and we mainly focused on non-apnea arousals that interrupted the sleep of an individual, including spontaneous arousals, as well as those triggered by hypopneas, apneas (central, obstructive and mixed), snores, or external stimuli (https://physionet.org/challenge/2018/). The final test dataset consists of 989 unseen polysomnographic recordings from different individuals. For each time point sampled at 200 Hz in each test sleep record, the participants needed to provide a prediction value between 0 and 1. A 8-h sleep record contained nearly 75 million data points ($8*60*60*200*13 = 74{,}880{,}000$). Our model made predictions for all the time points, at the resolution of 5 ms ($1/200 \, \text{Hz} = 5 \, \text{ms}$).

**Partition of the training, validation, and testing sleep records**. The 994 sleep records were randomly partitioned into three sets: 60% of them as the training set, 15% of them as the validation set, and 25% of them as the testing set. The validation set was used for monitoring the training-validation losses and avoiding the problems of overfitting or underfitting.

**Gaussian normalization**. The Gaussian normalization is calculated by

$$x_i' = (x_i - \bar{x})/s_x,$$

$$\bar{x} = \frac{1}{N}\sum_{i=1}^{N} x_i,$$

$$s_x = \frac{1}{N-1}\sum_{i=1}^{N}(x_i - \bar{x})^2,$$

where $x_i$ is the original value at time point $i$, $x_i'$ is the normalized value at time point $i$, and $N$ is the total number of time points. For the polysomnographic signals, we normalized each channel individually.

**Quantile normalization**. For each polysomnographic channel, we first ranked the original input vector

$$x_1, x_2, ..., x_N$$

into a sorted vector in the increasing order

$$x^1_{i1}, x^2_{i2}, ..., x^N_{iN},$$

where superscript number denotes the ranked increasing order, and the subscript number denotes the original position before ranking. Then we replace this sorted vector with a sorted reference vector

$$\text{ref}^1, \text{ref}^2, ..., \text{ref}^n,$$

which is also in increasing order. For example, $x^k_{ik}$ will be replaced by $\text{ref}^k$. Then we changed the order back and mapped $\text{ref}^k$ to its original position $ik$. After this

quantile normalization, the overall distribution of the input vector has been mapped to the distribution of the reference vector. The reference vector was pre-calculated by averaging all the sorted recordings in the training dataset. We quantile normalized each recording to the same reference to address potential batch and cohort effects. Each polysomnographic channel was normalized individually.

**AUROC and AUPRC**. Since sleep arousal events are extremely rare (<10% in terms of length), the performances of different methods are not apparent in the receiver operating characteristic (ROC) curve, where the $y$-axis is the true positive rate (TPR) and the $x$-axis is the false positive rate (FPR). The TPR and FPR are defined as

$$\text{TRP} = \text{TP}/(\text{TP} + \text{FN}),$$

$$\text{FPR} = \text{FP}/(\text{FP} + \text{TN}),$$

where TP is True Positive, FN is False Negative, FP is False Positive, and TN is True Negative. This is because when the number of negative events ("Sleep"; 92.8%), or TN, is much larger than the positive ones ("Arousal"; 7.2%), the FPR is always very small and will barely change even if a poor model makes many FP predictions. Therefore, in addition to the commonly used AUROC, we evaluated our model and various strategies using ARPRC[30,31]. In the Precision-Recall space, the Precision and Recall are defined as

$$\text{Precision} = \text{TP}/(\text{TP} + \text{FP}),$$

$$\text{Recall} = \text{TPR} = \text{TP}/(\text{TP} + \text{FN}).$$

The Precision is very sensitive to FP when the number of TP is relatively small. Therefore, the AUPRC metric is able to distinguish the performances in highly unbalanced data such as the annotations of sleep arousals.

**Convolutional neural network architectures**. The classic U-Net architecture was adapted in DeepSleep. The original U-Net is a 2D convolutional neural network designed for 2D image segmentation[18]. We transformed the structure into 1D for the time-series sleep records and largely increased the number of convolutional layers from the original 18 to 35 for extracting the information at different scales. Similar to U-Net, we had convolution, max pooling, and concatenation layers. The kernel size of 7 was used in the convolution operation and increasing the kernel size did not change the performance. The nonlinear activation after each convolution operation is a Rectified Linear Unit (ReLU) defined as

$$f(x) = \max(0, x),$$

where $x$ is the input to a neuron and $f(x)$ is the output. Only positive values activate a neuron and ReLU allows for fast and effective training of neural networks compared to other complex activation functions. In addition, batch normalization was used after each convolutional layer. In the final output layer, we used the sigmoid activation unit defined as

$$f(x) = \frac{1}{1 + e^{-x}},$$

where $x$ is the input to a neuron and $f(x)$ is the output. During the training process, the Adam optimizer was used with the learning rate of 1e−4 and the decay rate of 1e−5.

Other network structures were also tested, including long short-term memory (LSTM) and gated recurrent unit (GRU). They have similar performances. Therefore, we kept the U-Net based structure.

**Training losses**. The cross-entropy loss, or log loss, was used for model training in DeepSleep. The cross-entropy loss is defined as

$$H(y, \hat{y}) = \sum_{i=1}^{N}[-y_i \cdot \log \hat{y}_l - (1 - y_i) \cdot \log(1 - \hat{y}_l)],$$

where $y_i$ is the gold standard label of sleep = 0 or arousal = 1 at time point $i$, $\hat{y}_i$ is the prediction value at time point $i$, $N$ is the total number of time points, $y$ is the vector of the gold standard labels and $\hat{y}$ is the vector of predictions. Ideally, an "AUPRC loss" should be used for optimizing the prediction AUPRC. However, the "AUPRC loss" does not exist because the AUPRC function is not mathematically differentiable, which is required in the neural network model training through the back-propagation algorithm[32]. Therefore, we need to use cross-entropy loss to approximate the "AUPRC loss". Another option is using the Sorensen-dice coefficient defined as

$$S(y, \hat{y}) = \sum_{i=1}^{N}(y_i \cdot \hat{y}_l)/[\sum_{i=1}^{N}(y_i) + \sum_{i=1}^{N}(\hat{y}_l)],$$

where $y_i$ is the gold standard label of sleep = 0 or arousal = 1 at time point $i$, $\hat{y}_i$ is the prediction value at time point $i$, $N$ is the total number of time points, $y$ is the vector of the gold standard labels and $\hat{y}$ is the vector of predictions. We have tested

the cross-entropy loss, the Sorensen dice loss and combining these two losses. Using the cross-entropy loss achieved the best performance in DeepSleep.

**Overall AUPRC and AUROC.** The overall AUPRC, or the gross AUPRC, is defined as

$$\text{AUPRC} = \sum_j P_j(R_j - R_{j+1}),$$

$$P_j = \frac{\text{number of arousal data points with predicted probability } (j/1000) \text{ or greater}}{\text{total number of arousal data points with predicted probability } (j/1000) \text{ or greater}},$$

$$R_j = \frac{\text{number of arousal data points with predicted probability } (j/1000) \text{ or greater}}{\text{total number of arousal data points}},$$

where the Precision ($P_j$) and Recall ($R_j$) were calculated at each cutoff $j$ and $j = 0$, 0.001, 0.002, …, 0.998, 0.999, 1. For a test dataset of multiple sleep records, this overall AUPRC is similar to the "weighted AUPRC", which is different from simply averaging the AUPRC values of all test records. This is because the overall AUPRC considers the length of each record and longer records contributing more to the overall AUPRC, resulting in a more accurate performance description of a model. The overall AUPRC was also used as the primary scoring metric in the 2018 PhysioNet Challenge. The overall AUROC was defined in a similar way as the overall AUPRC.

**Validation on the SHHS datasets.** The large publicly available Sleep Heart Health Study (SHHS) dataset contains 6441 individuals in SHHS visit 1 (SHHS1) and 3295 individuals in SHHS visit 2 (SHHS2). The SHHS1 dataset was collected between 1995 and 1998, whereas the SHHS2 dataset was collected between 2001 and 2003. Since the recording montages were different among the PhysioNet, SHHS1, and SHHS2 datasets, the channels of polysomnograms were also different. For the SHHS1 and SHHS2 datasets, we only used a subset of 7 channels (SaO2, EEG-C3/A2, EEG-C4/A1, EOG-L, ECG, EMG, and Airflow), which were shared among these three datasets. In addition, the major signal sampling rates in the PhysioNet, SHHS1, and SHHS2 were 200, 125, and 250 Hz, respectively. We down-sample the signals to the same 25 Hz by averaging successive time points. Quantile normalization was used to address the potential cohort and batch effect. For both SHHS1 and SHHS2, we randomly selected 1000 recordings, which was comparable to the number of recordings ($n = 994$) in the PhysioNet training dataset. Then we applied DeepSleep pipeline to train, validate and test models on SHHS1 and SHHS2 datasets individually.

**Statistics and reproducibility.** For statistical analysis in this work, we performed Wilcoxon signed-rank test using R (version 3.6.1). In Fig. 2b, c, the statistical test was performed on the $n = 994$ samples of the PhysioNet dataset. In Fig. 4b, the statistical test was performed on the $n = 261$ test samples.

**Reporting summary.** Further information on research design is available in the Nature Research Reporting Summary linked to this article.

## Data availability

The datasets used in this study are publicly available at the 2018 PhysioNet Challenge website (https://physionet.org/physiobank/database/challenge/2018/)[33] and the Sleep Heart Health Study website (https://sleepdata.org/datasets/shhs)[34].

## Code availability

The code of DeepSleep is available at https://github.com/GuanLab/DeepSleep[35].

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

## Acknowledgements

This work is supported by NIH R35GM133346-01, NSF#1452656, and Michael J. Fox Foundation #17373 to YG. The work is also supported by 19AMTG34850176 (American

Heart Association and Amazon Web Services3.0 Data Grant Portfolio: Artificial Intelligence and Machine Learning Training Grants) to HL. We thank the GPU donation from Nvidia. We thank Ronald D. Chervin, MD, MS for input provided from the perspective of a sleep medicine physician.

## Author contributions

Y.G. and H.L. conceived and designed the winning algorithm in the "You Snooze, You Win" PhysioNet Challenge. H.Y. and Y.G. implemented the code of various neural network structures and augmentation strategies. HY performed post-challenge analyses. All authors contributed to the writing of the manuscript and approved the final manuscript.

## Competing interests

The authors declare no competing interests.
