## [Peer Review File · Communications Biology]

Reviewers' comments:

Reviewer #1 (Remarks to the Author):

The authors developed a new model for sleep arousal detection, winning an open competition, which demonstrates the superiority of the model. Applying deep models to large datasets is currently a popular approach, and extensive investigations / simulations presented in the supplemental material show the authors thoroughly investigated their model and its variations.

Having said this, after having studied the manuscript for a while, I remain with more questions than answers, mainly due to the lack understanding (1) why this model would be really better than other models (e.g. in the competition, second place had only 0.01 difference and this 0.01 difference is larger than the benefit of the fusion of multiple resolutions); (2) if this model is only good for arousal detection or if it could have a broader impact in sleep staging / annotation; (3) understanding what is the representation used after the encode block or what can be learned from it in terms of sleep /arousals.

Moreover, one has to study the supplementary material, which is not well described, to extract other important learnings, e.g. if all channels are needed, what is now the exact benefit of data augmentation.

In this spirit, having as title ... "resolves arousal detection" is a bit over claiming, and while I do like the architecture, I don't think the exhaustive list of AUPRC values do teach us anything about the specific architecture as competitor methods are not that exhaustively investigated neither (and probably most importantly) does it tell us anything about sleep arousals, and it is not mentioned what level of AUPRC is needed to really use it in clinical practice.

Reviewer #2 (Remarks to the Author):

This paper presents a novel approach for the segmentation of sleep arousal regions based on polysomnographic recordings. Long and short-range inter-dependencies are exploited to integrate information at different resolutions and scales for optimum performance as well as a data augmentation strategy based on channel swapping. The paper is well written and methods well presented, but I would like to see a bit more information provided.

- 1) Could the authors include some information about demographics for both PhysioNet/SHHS datasets in the text or as a Supplementary Table? It would be useful to know if these are healthy subjects/patients? Young/old? Were the training/testing/validation sets balanced in this regard?
- 2) Critically, would there be a difference in performance amongst these groups?
- 3) Rather than AUROC or AUPRC I wonder if a performance metric like the Jaccard Index(IoU) or Dice Coefficient would be more appropriate to evaluate overlap of the segmented arousal regions.
- 4) Fig 3A - if max pooling is used, please indicate this in the figure, and in the text on P6.
- 5) P9 - You mention that the AUPRC is correlated with the number of arousals in a sleep record. As a

control, can you also show that this is not correlated also with the length of the overall sleep record?

6) Traditionally, arousals are scored with just EEG and submental EMG measures. Would performance change if only these channels are used? In addition, if the channels are indeed similar and synchronized, perhaps it would also work with a very small subset of channels. This would greatly increase its utility in very portable EEG setups with minimal channels.

7) As with any DL algorithm, since we are comparing this to manually labelled data as the gold-standard, some of these 'false positives' detected by the algorithm could in fact be those missed by manual scorers. This, in addition to the point you raised about looking at longer segments of data, could indeed open a pathway for a re-review of these 'false positives' by sleep technologists and a better definition/understanding of arousals as a continuum, rather than a binary process.

All responses and edits in the manuscript are marked in Blue.

All original reviews are in Black.

Reviewer #1 :

The authors developed a new model for sleep arousal detection, winning an open competition, which demonstrates the superiority of the model. Applying deep models to large datasets is currently a popular approach, and extensive investigations / simulations presented in the supplemental material show the authors thoroughly investigated their model and its variations.

Thank you.

Having said this, after having studied the manuscript for a while, I remain with more questions than answers, mainly due to the lack understanding (1) why this model would be really better than other models (e.g. in the competition, second place had only 0.01 difference and this 0.01 difference is larger than the benefit of the fusion of multiple resolutions);

Thank you for communicating with us about this concern. Indeed it is commonly seen in benchmark studies that the top few are close, and some solutions might actually converge. Prior to this challenge, U-Net or similar type of structure was not identified to be a suitable method for this problem -- which we think is what we are contributing here. We applaud for the excellent performance of the second place (gapped by a difference of 0.55 versus 0.45 compared to the next one). It shares similarity in their methodology to ours, further corroborating the suitability of this model. As you said, claiming what 0.01 means is not a meaningful activity; we think what made this challenge and our work/method valuable is the group effort that brought forth the path that people can improve along. We have now added the descriptive comparison of the methodologies of the top methods:

Page 11:

“In fact, 8 out of the top 10 teams used neural network models in the PhysioNet Challenge (red blocks in **Fig. S2A**)¹⁶. Different input lengths and data preprocessing strategies were used, including raw polysomnogram data, features extracted by statistical analysis, short-time Fourier transform, or wavelet transform. ...In terms of input length, increasing input length significantly improved the performance, and full-length records were used by three teams (blue blocks in **Fig. S2A**). ...Deep learning approaches can model informative features in an implicit way without tedious feature crafting²⁴, and neural networks using raw data as input were frequently used by half of the top 10 teams (orange blocks in **Fig. S2A**).”

Fig. S2A Performance of top 10 teams in the PhysioNet Challenge. These methods are compared in terms of machine learning models (red blocks), input length for models (blue blocks), and the types of input (orange blocks)

(2) if this model is only good for arousal detection or if It could have a broader impact in sleep staging / annotation;

This original model was only good for arousal detection. But it could have a broader impact in sleep staging by small modifications.

In the revision, we trained a model using the sleep stage scores as the ground truth labels, based on the similar U-Net architecture. This is a multi-task deep learning model, which simultaneously scores REM (Rapid Eye Movement), N1, N2, N3, and Wake. We evaluated the predictive performance of our model using cross validations, which is excellent, and the results are now added to the revised manuscript as follows:

Page 13:

“Moreover, we extended our algorithm and trained a multi-task model using the sleep stage scores as the ground truth labels, including REM, N1, N2, N3, Wake. Similar to sleep arousal detection, we evaluated the predictive performance of our model using cross validations (**Fig. 4G-H**). In general, this multi-task model achieved high AUROCs and AUPRCs, demonstrating the robustness and generalizability of our deep learning pipeline in multiple sleep staging.”

Fig. 4G-H. We extended our method for sleep staging (REM, N1, N2, N3, and Wake) and evaluated performance using **(G)** AUPRCs and **(H)** AUROCs.

(3) understanding what is the representation used after the encode block or what can be learned from it in terms of sleep /arousals.

Thank you for this excellent suggestion! In the revision, we investigated the usage of the representations after the encoder blocks, which are the feature maps after the encoder layers. We treated the feature maps as input features to predict sleep-related characteristics, including Apnea-Hypopnea Index (AHI), percentages of sleep stages N1, N2, N3, and REM. We trained lightGBM models and found medium Pearson’s correlations between predictions and observations (**Fig. S7C**). These results indicate that these representations from encoder can be used to study the severity of sleep apnea, including AHI scores and percentages of different sleep stages. We have now added these results as follows:

Page 13:

“We further explored the potential usage of feature maps after the encoder blocks in our neural network. We trained a lightGBM model to predict other sleep characteristics, including apnea-hypopnea index (AHI) and percentages of N1/N2/N3/REM. We observed medium Pearson’s correlations between predictions and observations (**Fig. S7C**), indicating that the latent feature maps can be also used in sleep related studies.”

Fig. S7C. Other sleep characteristics, including apnea-hypopnea index (AHI) and percentages of N1/N2/N3/REM were also predicted using the lightGBM model based on the feature maps after the encoder blocks. The Pearson's correlations between predictions and observations are shown in (C).

Moreover, one has to study the supplementary material, which is not well described, to extract other important learnings, e.g. if all channels are needed, what is now the exact benefit of data augmentation.

We have now added two more figures to the main text instead of the supplementary material: (1) **Fig. 5** about input channels, data augmentation and normalization strategies, (2) **Fig. 7** about suspected sleep arousal detected by our algorithm but missed by manual scorers. For the 12 supplementary figures, we have thoroughly revised and added detailed descriptions about them in the main text.

Fig. 5. The performance comparison of models using different types of polysomnographic signals, augmentation strategies, normalization methods.

From left to right, the first six categories are EEG (channel 1-6), EOG (channel 7), EMG (channel 8-10), Airflow (channel 11), saturation of Oxygen (channel 12) and ECG (channel 13). The last one, “All”, represents the model using all these 13 channels as input. The prediction (A) AUPRCs and (B) AUROCs of models using different types of signals are shown in different colors. The model “All” using all 13 polysomnographic signals achieved the best performance. We further compared the prediction (C) AUPRCs and (D) AUROCs of different data augmentation strategies. The “Magnitude 1” strategy means that each training record was multiplied by a random number between 0.90 and 1.15, to simulate the fluctuation of the measurement in real life. The “Magnitude 2” strategy was the same as “Magnitude 1”, except for the range of the random number becoming wider, between 0.80 and 1.25. These two strategies had almost the same performance. The last “Magnitude+Length” strategy was built on top of “Magnitude 1”, in which we further extended or shrunk the record along the time axis by a random number between 0.90 and 1.15. This strategy decreased the performance and was not used in the final model training. In addition, the prediction (E) AUPRCs and (F) AUROCs of the Gaussian normalization and the quantile normalization were compared. In the Gaussian normalization, we first subtracted the average value of a signal then divided the signal by the standard deviation for each sleep record. In the quantile normalization, we first calculated the average of all training records as the reference record. Then for each record, we quantile normalized it to the reference record. The quantile normalization had better performance. We also compared the prediction (G) AUPRCs and (H) AUROCs of deep convolutional neural network (CNN) and logistic regression.

Fig. 7. A 180-second polysomnogram example with manual labels and predictions of sleep arousals. From top to bottom, the sleep arousal labels (arousal = 1 and non-arousal=0), predictions by our algorithm, and 13-channel polysomnograms are shown for a total of 180 seconds (six 30-second epochs). In addition to the two arousal events at the very beginning, our algorithm also detected two suspected sleep arousal events shown in dashed blue rectangles.

Detailed descriptions about supplementary figures:

Fig. S1 on Page 6:

“We tested input lengths ranging from 4,096 time points (~20 seconds) to 131,072 time points (~10 minutes), with single-channel input (**Fig. S1A-B**) or all-channel input (**Fig. S1C-D**). Intriguingly, when we trained the convolutional neural networks on longer sleep records, we consistently achieved better performances. Therefore, we used the entire sleep record as input to make predictions, instead of small segments of a sleep record.”

Fig. S2 on Page 11:

“In fact, 8 out of the top 10 teams used neural network models in the PhysioNet Challenge (red blocks in **Fig. S2A**)¹⁶. Different input lengths and data preprocessing strategies were used, including raw polysomnogram data, features extracted by statistical analysis, short-time Fourier transform, or wavelet transform. ... In terms of input length, increasing input length significantly improved the performance, and full-length records were used by three teams (blue blocks in **Fig. S2A**).”

“Deep learning approaches can model informative features in an implicit way without tedious feature crafting²⁴, and neural networks using raw data as input were frequently used by half of the top 10 teams (orange blocks in **Fig. S2A**).”

Fig. S3 on Page 12:

“To comprehensively investigate the effects of various network structures and parameters on predictions, we further performed experiments with different modifications, including (1) the “shallow” neural network with less convolutional layers (**Fig. S3A-B**), (2) using average pooling instead of max pooling (**Fig. S3C-D**), (3) larger convolution kernel size from 7 to 11 (**Fig. S3E-F**), and (4) using the Sorensen dice loss function instead of cross-entropy loss (**Fig. S3G-H**). These modifications had either similar or lower prediction performances.”

Fig. S4 on Page 12:

“Subsequent analysis of the relationships between the predictive performance and the number of arousals were performed (**Fig. S4A-B**). As we expected, the prediction AUPRC was positively correlated with the number of arousals in a sleep record. The individuals who had more sleep arousals during sleep were relatively easier to predict. As a control, we also calculated Pearson's correlations between AUPRC/AUROC and the total length of sleep record (**Fig. S4C-D**). These correlations are close to zeros and not statistically significant, with the p -values of 0.176 and 0.316 respectively. Moreover, we tested the runtime of DeepSleep with Graphics Processing Unit (GPU) acceleration and segmenting sleep arousals of a full sleep record can be finished within 10 seconds on average (**Fig. S4E-F**).”

Fig. S5 on Page 12:

“We observed similar performances of detecting sleep arousals on the PhysioNet, SHHS1, and SHHS2 datasets in **Fig. S5A-B**, demonstrating the robustness of our method. We also summarized the available clinical characteristics of the PhysioNet, SHHS1, and SHHS2 datasets in **Table S2-4**, including gender, age, race and disease. The different groups were balanced for the training and testing subsets. We further evaluated the predictive performance for these groups (**Fig. S5C-H** and **Fig. S6**). In general, the performances were very similar among different groups with the AUPRCs around 0.6. We found slightly higher performance for males than females in all three datasets (**Fig. S5C-E**). There are some differences for different age groups, but no clear trends were observed (**Fig. S5F-H**).”

Fig. S6 on Page 12:

“In terms of race, we found slightly higher predictive performance for the others and white groups than the black group (**Fig. S6A-B**). For the SHHS1 and SHHS2 datasets, we also considered patients with different cardiovascular diseases/events, including myocardial infarction (MI), stroke, angina, and congestive heart failure (CHF). The predictive performances of different groups were similar, with the AUPRCs ranging from 0.56 to 0.61 (**Fig. S6C-D**).”

Fig. S7 on Page 13:

“Specifically, we applied DeepSleep pipeline to the PhysioNet data and built three types of models for detecting (1) apneic arousals; (2) non-apneic arousals; and (3) all arousals (apneic and non-apneic arousals). DeepSleep is able to detect both apneic and non-apneic arousals (**Fig. S7A-B**). ... We further explored the potential usage of feature maps after the encoder blocks in our neural network. We trained a lightGBM model to predict other sleep characteristics, including apnea-hypopnea index (AHI) and percentages of N1/N2/N3/REM. We observed medium Pearson’s correlations between predictions and observations (**Fig. S7C**), indicating that the latent feature maps can be also used in sleep related studies.”

Fig. S8 on Page 14:

“An example 7.5-hour sleep record with the prediction AUROC of 0.960 and AUPRC of 0.761 is shown in **Fig. 6**. More examples at 3 rank percentiles (25%, 50%, and 75%) based on the AUPRC values can be found in **Fig. S8**.”

Fig. S9-12 on Page 15:

“It is critical to compare predictions of our algorithm and the ground truth labels created by sleep scorers, since the manual scores may not be perfect. Specifically, we focused on the false positives detected by our method and added five examples from five different individuals in **Fig. 7** and **Fig. S9-12**. In each figure, the ground truth arousal labels created by human scorers (arousal=1 and non-arousal=0) and our predictions are shown in the top two rows. The 13 physiological channels from the polysomnogram are also shown below. A total of six 30-second epochs are shown in each figure, containing multiple true positives and false positives (dashed blue rectangles). For example, in **Fig. 7** (tr04-0933), there are two sleep arousal events detected by both manual scorers and our algorithm. Our algorithm also found two other suspected sleep arousal events shown in dashed blue rectangles. By examining the corresponding polysomnographic signals, we found that these two false positives were likely to be sleep arousals missed by the scorers - there are sudden shifts in the EEG and EOG channels, and suspected respiratory effort related changes in the EMG channels (ABD and Chest) and Airflow. Similarly, we identified multiple false positives for other individuals in **Fig. S9-12**. These results indicate that our computational method can potentially detect suspected arousals, complementing the scoring by human experts and highlighting the regions of interest to assist sleep scoring.”

In this spirit, having as title ... "resolves arousal detection" is a bit over claiming, and while I do like the architecture, I don t think the exhaustive list of AUPRC values do teach us anything about the specific architecture as competitor methods are not that exhaustively investigated neither (and probably most

importantly) does it tell us anything about sleep arousals, and it is not mentioned what level of AUPRC is needed to really use it in clinical practice.

We have now revised the title as “DeepSleep: accurate and fast detection of sleep arousal by deep convolutional neural network” (Page 1).

Related to how AUPRC values can teach us anything about sleep arousal, it is indeed not directly related. But we thought it would be good to include some of the biological insights from the models, and what actually we could have helped the clinicians with. Thus, in the revised manuscript, we have investigated “false positives” reported by our algorithm and find that many cases are suspected sleep arousals missed by manual scorers, which are supported by the patterns and frequency changes in the EEG, EOG, EMG and Airflow channels (**Fig. 7** and **S9-12** below). Thus we think one of the usages is actually complementing and assisting the current manual sleep scoring process and potentially improving the scoring criteria by examining the wrongly scored cases. This has been added on Page 15:

“It is critical to compare predictions of our algorithm and the ground truth labels created by sleep scorers, since the manual scores may not be perfect. Specifically, we focused on the false positives detected by our method and added five examples from five different individuals in **Fig. 7** and **Fig. S9-12**. In each figure, the ground truth arousal labels created by human scorers (arousal=1 and non-arousal=0) and our predictions are shown in the top two rows. The 13 physiological channels from the polysomnogram are also shown below. A total of six 30-second epochs are shown in each figure, containing multiple true positives and false positives (dashed blue rectangles). For example, in **Fig. 7** (tr04-0933), there are two sleep arousal events detected by both manual scorers and our algorithm. Our algorithm also found two other suspected sleep arousal events shown in dashed blue rectangles. By examining the corresponding polysomnographic signals, we found that these two false positives were likely to be sleep arousals missed by the scorers - there are sudden shifts in the EEG and EOG channels, and suspected respiratory effort related changes in the EMG channels (ABD and Chest) and Airflow. Similarly, we identified multiple false positives for other individuals in **Fig. S9-12**. These results indicate that our computational method can potentially detect suspected arousals, complementing the scoring by human experts and highlighting the regions of interest to assist sleep scoring.”

Fig. 7. A 180-second polysomnogram example with manual labels and predictions of sleep arousals. From top to bottom, the sleep arousal labels (arousal = 1 and non-arousal=0), predictions by our algorithm, and 13-channel polysomnograms are shown for a total of 180 seconds (six 30-second epochs). In addition to the two arousal events at the very beginning, our algorithm also detected two suspected sleep arousal events shown in dashed blue rectangles.

Fig. S9. A 180-second polysomnogram example with manual labels and predictions of sleep arousals.

From top to bottom, the sleep arousal labels (arousal = 1 and non-arousal=0), predictions by our algorithm, and 13-channel polysomnograms are shown for a total of 180 seconds (six 30-second epochs). Our algorithm detected suspected sleep arousal events shown in dashed blue rectangles.

Fig. S10. A 180-second polysomnogram example with manual labels and predictions of sleep arousals.

From top to bottom, the sleep arousal labels (arousal = 1 and non-arousal=0), predictions by our algorithm, and 13-channel polysomnograms are shown for a total of 180 seconds (six 30-second epochs). Our algorithm detected suspected sleep arousal events shown in dashed blue rectangles.

Fig. S11. A 180-second polysomnogram example with manual labels and predictions of sleep arousals.

From top to bottom, the sleep arousal labels (arousal = 1 and non-arousal=0), predictions by our algorithm, and 13-channel polysomnograms are shown for a total of 180 seconds (six 30-second epochs). Our algorithm detected suspected sleep arousal events shown in dashed blue rectangles.

Fig. S12. A 180-second polysomnogram example with manual labels and predictions of sleep arousals.

From top to bottom, the sleep arousal labels (arousal = 1 and non-arousal=0), predictions by our algorithm, and 13-channel polysomnograms are shown for a total of 180 seconds (six 30-second epochs). Our algorithm detected suspected sleep arousal events shown in dashed blue rectangles.

We hope the above has cleared your worries and we hope with the edits in response to you and the other reviewer, the article has become more amiable. Thank you.

Reviewer #2 :

This paper presents a novel approach for the segmentation of sleep arousal regions based on polysomnographic recordings. Long and short-range inter-dependencies are exploited to integrate information at different resolutions and scales for optimum performance as well as a data augmentation strategy based on channel swapping. The paper is well written and methods well presented, but I would like to see a bit more information provided.

Thank you.

1) Could the authors include some information about demographics for both PhysioNet/SHHS datasets in the text or as a Supplementary Table? It would be useful to know if these are healthy subjects/patients? Young/old? Were the training/testing/validation sets balanced in this regard?

Thank you for pointing this out. We have added the available demographic information in the revised manuscript as Supplementary Table S2-4 for the PhysionNet, SHHS1, and SHHS2 datasets. The different groups are balanced in the total/training/testing sets. More descriptions about this have been added to the revised manuscript on Page 12:

“We also summarized the available clinical characteristics of the PhysioNet, SHHS1, and SHHS2 datasets in **Table S2-4**, including gender, age, race and disease. The different groups were balanced for the training and testing subsets.”

Table S2

clinical characteristics of the 2018 PhysioNet Challenge dataset			
	total (n=1893)	train (n=994)	test (n=989)
gender (% male)	65	67	63
average age	55	55	55
drug use (%)			
antidepressant	26.1	25.7	26.5
antihistamine	4.8	4.8	4.8
benzodiazepine	16.1	16.9	15.4
hypertension	40.9	41	40.6

neuroleptic	4.2	4.5	3.8
opiate	7.4	8.1	6.7
neuroactive	19.1	20.8	17.5
sleep aids	28.3	29	27.8

Table S3

clinical characteristics of the SHHS Visit1 dataset			
	total (n=1000)	train (n=750)	test (n=250)
gender (% male)	43.0	44.0	40.0
average age	72.8	72.9	72.5
race (%)			
white	85.4	85.5	85.2
black	11.5	12.1	9.6
others	3.1	2.4	5.2
disease (%)			
myocardial infarction	22.6	22.9	21.6
stroke	17.0	16.3	19.2
angina	18.1	19.3	14.4
congestive heart failure	23.8	23.3	25.2

Table S4

clinical characteristics of the SHHS Visit2 dataset			
	total (n=1000)	train (n=750)	test (n=250)
gender (% male)	44.0	43.7	44.8
average age	67.9	68.2	66.9
race (%)			

white	86.1	85.3	88.4
black	11.0	11.6	9.2
others	2.9	3.1	2.4
disease (%)			
myocardial infarction	17.3	18.3	14.4
stroke	8.2	8.0	8.8
angina	31.2	30.4	33.6
congestive heart failure	16.3	16.5	15.6

2) Critically, would there be a difference in performance amongst these groups?

In this revision, we evaluated the performance for different groups in terms of gender, age, race, and disease for three datasets. In general, the performances are very similar among different groups. We found slightly higher performance for males than females in all three datasets (**Fig. S5C-E**). There are some differences for different age groups, but no clear trends were observed (**Fig. S5F-H**). In terms of race, we found slightly higher predictive performance for the others and white groups than the black group (**Fig. S6A-B**). For the SHHS1 and SHHS2 datasets from the Sleep Heart Health Study, we also consider patients with cardiovascular diseases/events, including myocardial infarction (MI), stroke, angina and congestive heart failure (CHF). The predictive performances of different groups were similar, with the AUPRCs ranging from 0.56 to 0.61 (**Fig. S6C-D**). These results are now added to the revised manuscript on Page 12:

“We further evaluated the predictive performance for these groups (**Fig. S5C-H** and **Fig. S6**). In general, the performances were very similar among different groups with the AUPRCs around 0.6. We found slightly higher performance for males than females in all three datasets (**Fig. S5C-E**). There are some differences for different age groups, but no clear trends were observed (**Fig. S5F-H**). In terms of race, we found slightly higher predictive performance for the others and white groups than the black group (**Fig. S6A-B**). For the SHHS1 and SHHS2 datasets, we also considered patients with different cardiovascular diseases/events, including myocardial infarction (MI), stroke, angina, and congestive heart failure (CHF). The predictive performances of different groups were similar, with the AUPRCs ranging from 0.56 to 0.61 (**Fig. S6C-D**).”

Fig. S5C-H. The performance comparison of DeepSleep on three different datasets and different groups of individuals.

Fig. S6. The performance comparison of DeepSleep on individuals of different races (**A-B**) and cardiovascular conditions (**C-D**).

3) Rather than AUROC or AUPRC I wonder if a performance metric like the Jaccard Index(IoU) or Dice Coefficient would be more appropriate to evaluate overlap of the segmented arousal regions.

Thank you for this suggestion. In addition to AUPRC and AUROC (**Fig. 4C-D**), we have used the Jaccard Index and Sorensen Dice Coefficient as two extra performance measures (**Fig. 4E-F**). We found similar trends using these four different metrics and the ensemble model of three resolutions consistently achieved the best performance (see below).

Page 9:

“This channel swapping strategy was bold but effective, adapting which largely improved the prediction performance (“1/8_no_swap” versus “1/8” evaluated by the AUPRC, AUROC, Sorensen dice coefficient, and Jaccard index in **Fig. 4C-F**). Finally, we assembled the predictions from the “1/8”, “1/2” and “full” resolution models as the final prediction in DeepSleep (“1/8+1/2+full” in **Fig. 4C-F**.)”

Fig. 4C-F. The prediction (C) AUPRCs, (D) AUROCs, (E) Sorensen dice coefficient, and (F) Jaccard index of models using different resolutions or strategies were calculated. The “1/8_no_swap” model corresponds to the model using the “1/8” resolution records as input without any channel swapping, whereas the “1/8”, “1/2” and “full” models use the strategy of swapping similar polysomnographic channels. The final “1/8+1/2+full” model of DeepSleep is the ensemble of models at 3 different resolutions, achieving the highest AUPRC of 0.550 and AUROC of 0.927.

4) Fig 3A - if max pooling is used, please indicate this in the figure, and in the text on P6.

Thank you. We have added this information in Fig 3B now.

Fig. 3. The deep convolutional neural network architecture in DeepSleep.

5) P9 - You mention that the AUPRC is correlated with the number of arousals in a sleep record. As a control, can you also show that this is not correlated also with the length of the overall sleep record?

Yes, we have added **Fig. S4C-D** of (1) AUPRCs VS lengths of sleep records, and (2) AUROCs VS lengths of sleep records. As we expected, Pearson's correlations are close to zero. These correlations are not statistically significant, with the *p*-values of 0.176 and 0.316 respectively.

Page 12:

“As we expected, the prediction AUPRC was positively correlated with the number of arousals in a sleep record. The individuals who had more sleep arousals during sleep were relatively easier to predict. As a control, we also calculated Pearson's correlations between AUPRC/AUROC and the total length of sleep record (**Fig. S4C-D**). These correlations are close to zeros and not statistically significant, with the *p*-values of 0.176 and 0.316 respectively.”

Fig. S4C-D

6) Traditionally, arousals are scored with just EEG and submental EMG measures. Would performance change if only these channels are used? In addition, if the channels are indeed similar and synchronized, perhaps it would also work with a very small subset of channels. This would greatly increase its utility in very portable EEG setups with minimal channels.

To test if a small subset of channels is sufficient to detect sleep arousal, we benchmark the performance of (1) models using all channels, and (2) models using one type of signals (EEG, EOG, EMG, Airflow, Saturation of Oxygen, or ECG). The results are shown in **Fig. 5A-B**. We found that models with only EMG achieved relatively high performance (AUPRC=0.476, AUROC=0.902), close to the model with all channels (AUPRC=0.520, AUROC=0.919). For models with other types of channels, predictive AUPRCs are around 0.3 and AUROCs are around 0.8. These results indicate that using a subset of channels can also effectively predict sleep arousals, which can be used for portable/in-home facilities with minimal channels. These results are shown on Page 11:

“We further investigated which types of physiological signals are necessary for sleep arousal detections and benchmarked the performance of (1) models using all channels, and (2) models using one type of signals (EEG, EOG, EMG, Airflow, Saturation of Oxygen, or ECG). The results are shown in **Fig. 5A-B**. We found that models with only EMG achieved relatively high performance (AUPRC=0.476, AUROC=0.902), close to the model with all channels (AUPRC=0.520, AUROC=0.919). For models with other types of channels, the AUPRCs and AUROCs are around 0.3 and 0.8, respectively. The 13 polysomnographic channels complemented each other and using all of them instead of one type of signals allowed the neural network to capture interactions between channels and achieved the highest performance.”

Fig. 5A-B. From left to right, the first six categories are EEG (channel 1-6), EOG (channel 7), EMG (channel 8-10), Airflow (channel 11), saturation of Oxygen (channel 12) and ECG (channel 13). The last one, “All”, represents the model using all these 13 channels as input. The prediction (A) AUPRCs and (B) AUROCs of models using different types of signals are shown in different colors.

7) As with any DL algorithm, since we are comparing this to manually labelled data as the gold-standard, some of these 'false positives' detected by the algorithm could in fact be those missed by manual scorers. This, in addition to the point you raised about looking at longer segments of data, could indeed open a pathway for a re-review of these 'false positives' by sleep technologists and a better definition/understanding of arousals as a continuum, rather than a binary process.

Thank you for this great suggestion! In this revision, we investigated the false positives detected by our method and added five examples from five different individuals in **Fig. 7** and **Fig. S9-12** (see below). In each figure, the ground truth arousal labels created by human scorers (arousal=1 and non-arousal=0) and our predictions are shown in the top two rows. The 13 physiological channels from the polysomnogram are also shown below. A total of six 30-second epochs are shown in each figure, containing multiple true positives and false positives (dashed blue rectangles). For example, in **Fig. 7**(tr04-0933), there are two sleep arousal events detected by both manual scorers and our algorithm. Our algorithm also found two other suspected sleep arousal events shown in dashed blue rectangles. By examining the corresponding polysomnographic signals, we found that these two false positives are likely to be sleep arousals missed by the scorers. For example, there are sudden shifts in the EEG and EOG channels, and suspected respiratory effort related changes in the EMG channels (ABD and Chest) and Airflow. Similarly, we identified multiple false positives in **Fig. S9-12**. These results indicate that our computational method can

potentially detect suspected arousals, complementing the scoring by human experts and highlighting the regions of interest to assist sleep scoring. We have now added this results to the revised manuscript on Page 15:

“It is critical to compare predictions of our algorithm and the ground truth labels created by sleep scorers, since the manual scores may not be perfect. Specifically, we focused on the false positives detected by our method and added five examples from five different individuals in **Fig. 7** and **Fig. S9-12**. In each figure, the ground truth arousal labels created by human scorers (arousal=1 and non-arousal=0) and our predictions are shown in the top two rows. The 13 physiological channels from the polysomnogram are also shown below. A total of six 30-second epochs are shown in each figure, containing multiple true positives and false positives (dashed blue rectangles). For example, in **Fig. 7** (tr04-0933), there are two sleep arousal events detected by both manual scorers and our algorithm. Our algorithm also found two other suspected sleep arousal events shown in dashed blue rectangles. By examining the corresponding polysomnographic signals, we found that these two false positives were likely to be sleep arousals missed by the scorers - there are sudden shifts in the EEG and EOG channels, and suspected respiratory effort related changes in the EMG channels (ABD and Chest) and Airflow. Similarly, we identified multiple false positives for other individuals in **Fig. S9-12**. These results indicate that our computational method can potentially detect suspected arousals, complementing the scoring by human experts and highlighting the regions of interest to assist sleep scoring.”

Fig. 7. A 180-second polysomnogram example with manual labels and predictions of sleep arousals. From top to bottom, the sleep arousal labels (arousal = 1 and non-arousal=0), predictions by our algorithm, and 13-channel polysomnograms are shown for a total of 180 seconds (six 30-second epochs). In addition to the two arousal events at the very beginning, our algorithm also detected two suspected sleep arousal events shown in dashed blue rectangles.

Fig. S9. A 180-second polysomnogram example with manual labels and predictions of sleep arousals.

From top to bottom, the sleep arousal labels (arousal = 1 and non-arousal=0), predictions by our algorithm, and 13-channel polysomnograms are shown for a total of 180 seconds (six 30-second epochs). Our algorithm detected suspected sleep arousal events shown in dashed blue rectangles.

Fig. S10. A 180-second polysomnogram example with manual labels and predictions of sleep arousals.

From top to bottom, the sleep arousal labels (arousal = 1 and non-arousal=0), predictions by our algorithm, and 13-channel polysomnograms are shown for a total of 180 seconds (six 30-second epochs). Our algorithm detected suspected sleep arousal events shown in dashed blue rectangles.

Fig. S11. A 180-second polysomnogram example with manual labels and predictions of sleep arousals.

From top to bottom, the sleep arousal labels (arousal = 1 and non-arousal=0), predictions by our algorithm, and 13-channel polysomnograms are shown for a total of 180 seconds (six 30-second epochs). Our algorithm detected suspected sleep arousal events shown in dashed blue rectangles.

Fig. S12. A 180-second polysomnogram example with manual labels and predictions of sleep arousals.

From top to bottom, the sleep arousal labels (arousal = 1 and non-arousal=0), predictions by our algorithm, and 13-channel polysomnograms are shown for a total of 180 seconds (six 30-second epochs). Our algorithm detected suspected sleep arousal events shown in dashed blue rectangles.

REVIEWERS' COMMENTS:

Reviewer #1 (Remarks to the Author):

I think this became a very nice and detailed piece of work, and I recommend acceptance in its current form.

Reviewer #2 (Remarks to the Author):

Thank you for the thorough response to my queries.

As a follow up to point 2, can the authors discuss why performance is poorer in females vs. males, and blacks compared to white/others? One reason could be that there were a lot fewer blacks in the sample, but then again there were even fewer of those in the 'Others' category. What about the actual distribution of arousals between males/females and whites/blacks/others? This could help us understand the improved performance in certain groups compared to others since there is a relationship between AUPRC and number of arousals in a record. Can a binary process.

All original reviews are in Black.

All responses and edits in the manuscript are marked in Blue.

Reviewer #1 (Remarks to the Author):

I think this became a very nice and detailed piece of work, and I recommend acceptance in its current form.

Thank you!

Reviewer #2 (Remarks to the Author):

Thank you for the thorough response to my queries.

As a follow up to point 2, can the authors discuss why performance is poorer in females vs. males, and blacks compared to white/others? One reason could be that there were a lot fewer blacks in the sample, but then again there were even fewer of those in the 'Others' category. What about the actual distribution of arousals between males/females and whites/blacks/others? This could help us understand the improved performance in certain groups compared to others since there is a relationship between AUPRC and number of arousals in a record.

Thank you for this suggestion. We calculated the AUPRC baselines of samples in different gender/race groups on the Physionet, SHHS1 and SHHS2 datasets. The main reason is that the AUPRC baselines of different groups were different. In general, the group with a higher baseline has higher predictive performance. We have added Supplementary Table S5 and discussion about these results on Page 7:

“The prediction AUPRC is highly associated with the AUPRC baseline, which is the ratio of sleep arousal time over the total sleep time. For different gender and race groups, the main reason for a higher AUPRC was that the corresponding baseline was higher (**Supplementary Table 5**). For example, on average males commonly had more sleep arousals (higher AUPRC baselines) than females in three datasets. As we expected, our model achieved higher AUPRCs for males.”

Supplementary Table 5. Average AUPRC baseline and predictive performance for different gender and race groups

	baseline	prediction
gender		
male (Physionet)	0.077	0.509
female (Physionet)	0.061	0.475

male (SHHS1)	0.050	0.603
female (SHHS1)	0.040	0.598
male (SHHS2)	0.044	0.619
female (SHHS2)	0.032	0.600
race		
white (SHHS1)	0.044	0.601
black (SHHS1)	0.035	0.574
others (SHHS1)	0.065	0.627
white (SHHS2)	0.037	0.609
black (SHHS2)	0.045	0.585
others (SHHS2)	0.049	0.698